# A Decoupled Semantic–Detail Learning Network for Remote Sensing Object Detection in Complex Backgrounds

Hao Ruan(), Wenbin Qian *, Zhihong Zheng() and Yingqiong Peng

School of Software, Jiangxi Agricultural University, Nanchang 330045, China; ruanhao@stu.jxau.edu.cn (H.R.); zhengzhihong@stu.jxau.edu.cn (Z.Z.); jneyq@jxau.edu.cn (Y.P.)
* Correspondence: qianwenbin@jxau.edu.cn

**Abstract:** Detecting multi-scale objects in complex backgrounds is a crucial challenge in remote sensing. The main challenge is that the localization and identification of objects in complex backgrounds can be inaccurate. To address this issue, a decoupled semantic–detail learning network (DSDL-Net) was proposed. Our proposed approach comprises two components. Firstly, we introduce a multi-receptive field feature fusion and detail mining (MRF-DM) module, which learns higher semantic-level representations by fusing multi-scale receptive fields. Subsequently, it uses multi-scale pooling to preserve detail texture information at different scales. Secondly, we present an adaptive cross-level semantic–detail fusion (CSDF) network that leverages a feature pyramid with fusion between detailed features extracted from the backbone network and high-level semantic features obtained from the topmost layer of the pyramid. The fusion is accomplished through two rounds of parallel global–local contextual feature extraction, with shared learning for global context information between the two rounds. Furthermore, to effectively enhance fine-grained texture features conducive to object localization and features conducive to object semantic recognition, we adopt and improve two enhancement modules with attention mechanisms, making them simpler and more lightweight. Our experimental results demonstrate that our approach outperforms 12 benchmark models on three publicly available remote sensing datasets (DIOR, HRRSD, and RSOD) regarding average precision (AP) at small, medium, and large scales. On the DIOR dataset, our model achieved a 2.19% improvement in $mAP$@0.5 compared to the baseline model, with a parameter reduction of 14.07%.

**Keywords:** detailed feature; feature fusion; multi-scale receptive fields; remote sensing datasets; semantic–detail learning

## 1. Introduction

Object detection in remote sensing, including the detection of aerial and satellite images, has significant practical applications in various fields, such as urban planning, land use, natural resource management, and disaster response [1–4]. In the early stages of remote sensing object detection, traditional methods based on handcrafted features were used, but their performance was limited. With the emergence of deep learning, these methods have gradually been replaced by deep learning-based algorithms in remote sensing object detection, which can generally be categorized into three types: one-stage, two-stage, and anchor-free methods. One-stage methods, such as the YOLO series [5,6], directly predict both the object and its bounding box after scanning an image once. These types of models use anchor boxes to regress the object's coordinate position and adopt multi-scale feature fusion to alleviate the problem of insufficient feature learning. This allows the network to learn higher semantic levels, resulting in improved accuracy. Such networks are known for their fast inference time and are widely used in industry. However, due to the simplicity of their prediction method and their smaller model size, their accuracy is lower compared to other types of object detection algorithms. Two-stage methods, such as the R-CNN series [7–10], divide the detection task into two stages. In the first stage, a

region proposal network is employed to generate high-quality candidate regions, similar to a simple pre-prediction process. This significantly improves the quality of positive samples compared to one-stage models using preset anchor boxes. In the second stage, the model classifies and predicts the refined bounding box for the generated candidate regions. These types of models are larger and generally slower than one-stage models due to the two-stage prediction process. Nevertheless, they exhibit higher robustness and accuracy. Finally, anchor-free methods, such as RepPoints [11], predict both the object and its bounding box directly without using anchor boxes. This is typically achieved by dense sampling or similar convolution-like operations to predict the object's position and size. This enables them to better adapt to objects with different shapes and sizes. Research has shown that deep learning-based remote sensing object detection is highly effective in a variety of applications. Despite significant progress in remote sensing object detection, accurately detecting and localizing objects in complex background scenes with little difference between the object and its surroundings remains a challenge [12]. Overcoming this challenge is crucial for advancing modern high-tech applications, including but not limited to unmanned aerial vehicles, precision agriculture, rescue operations, and environmental survey tasks [13–16]. However, the existing algorithms tackling this problem face three limitations: (1) an insufficient semantic hierarchy of extracted features leading to detection errors, (2) the coupling interaction between details and semantics during feature extraction leads to the loss of detailed features, and (3) the inability to fully utilize high-level semantic features in conjunction with fine-grained details for accurate object localization. To address these limitations, a decoupled semantic–detail learning network (DSDL-Net) was proposed to tackle this problem. The main contributions of this paper are as follows.

1. For the issues of insufficient semantic hierarchy and detail loss in detection networks, we propose an MRF-DM module that maintains detailed information while producing higher-level semantic features.
2. For the issues of ineffective integration between detail texture information and high-level semantic features, we propose an adaptive cross-level semantic–detail fusion (CSDF) network that effectively integrates almost lossless detail information without compromising learned semantic features.
3. Multiple experiments, including ablation experiments on three remote sensing object detection datasets (DIOR, HRRSD, and RSOD) and comparative experiments on the DIOR dataset, were conducted to validate the performance of the model.

## 2. Related Work

Detecting objects against complex backgrounds remains a persistent challenge in the field, and there has been extensive research exploring various approaches to tackle this problem. In this section, we will review the representative related work on feature fusion, collaborative learning of semantics and details, and attention mechanisms.

### 2.1. Feature Fusion

In cases where the semantic hierarchy is insufficient to distinguish between objects and backgrounds, or where the lack of detail information hinders the network's ability to accurately locate objects, feature fusion is a widely adopted solution. By employing a variety of techniques to increase the semantic hierarchy and enrich detail information, feature fusion enhances the expressive power of features, thereby improving object detection and localization. The feature pyramid network (FPN), originally designed by Lin et al. [17], adopts a top–down architecture with lateral connections to effectively fuse multi-scale features, combining high-level semantic features with lower-level features to increase the contextual information for object recognition and enhance the representation of semantic features. Mei et al. [18] proposed the path aggregation network (PANet), which further enhances the semantic level of features by adding a bottom-to-up feature fusion path to the FPN. Subsequently, more complex and refined feature fusion methods have been proposed, such as Tan et al.'s [19] EfficientDet network which uses a feature fusion method called

Bi-FPN, which employs weighted sum and residual connections to adjust the contribution of each scale feature map. This enhances feature representation while reducing network complexity by removing the fusion of single inputs without affecting performance. While most mainstream object detection methods previously relied on multi-level feature fusion achieved through methods such as multi-scale feature joint, Liu et al. [20] proposed an adaptive learning spatial weight method named adaptively spatial feature fusion (ASFF) to fuse multi-scale feature maps. Later, Zha et al. [21] proposed a feature fusion method based on the BA block, which combines the Bi-FPN and ASFF methods to significantly improve network robustness.

### 2.2. Collaborative Learning of Semantic and Detailed Information

Semantic features provide high-level information about the object's class and attributes, while detail features provide low-level information about the object's shape, texture, and other visual characteristics. Both types of features are crucial for accurately recognizing objects in images. However, the interaction between details and semantics during the learning process in convolutional neural networks (CNNs) can lead to a loss of detailed information as the semantic hierarchy ascends during network feature extraction. Additionally, because remote sensing images are usually large, multiple pooling operations can result in a significant loss of object details. These challenges have made it difficult to effectively facilitate collaborative learning between semantics and details. To address this, several studies have been conducted. Zha et al. [22] proposed a four-scale residual feature fusion network to obtain detailed features, which are then effectively fused with high semantic output features from the feature pyramid. Jiang et al. [23] proposed a spatial semantic joint context method to fuse detail and semantic information by utilizing detailed spatial cues contained in a multi-scale local context and generalized semantic information encoded in a global context, enhancing the feature expression of objects. Liang et al. [24] proposed a parallel decoupling learning approach to separately extract high semantic and rich detail features, allowing for the effective fusion of both types of information to improve the accuracy of saliency detection. Zhou et al. [25] proposed a full-scale feature fusion siamese network (F3SNet) that enhances the spatial localization of deep features by densely connecting raw image features from shallow to deep layers and complements the changing semantics of shallow features by densely connecting the concatenated feature maps from deep to shallow layers. Yu et al. [26] introduced boundary information to perform salient object detection in optical remote sensing images (RSIs) by combining the encoder's low-level and high-level features via a feature-interaction operation, yielding boundary information, and then introducing the boundary cues into each decoder block to focus more on the boundary details and objects simultaneously.

### 2.3. Attention Mechanism

Attention mechanisms can be incorporated into convolutional neural networks (CNNs) as a flexible module for feature learning. They enable the network to selectively attend to important features while suppressing irrelevant ones. Additionally, attention mechanisms can effectively learn or emphasize global semantic and local detail features and can be positioned at various points within the network to enhance learning. For example, Squeeze-and-Excitation Networks (SE-Net), first introduced by Hu et al. [27], learn attention weights to capture global semantic information from each channel of the feature. This can improve the network's recognition ability in complex background scenes. SimAM, introduced by Hu et al. [28], generates 3D weights for each feature without using any parameters. This allows the adjustment of both global semantic information and local detail features within the feature layer simultaneously. This attention mechanism can effectively coordinate the learning of semantics and details. Additionally, Pan et al.'s [29] ACMix attention module unifies convolutional and self-attention paradigms, endowing the convolution with self-attention mechanism characteristics, which can effectively mine intrinsic detail information in feature maps during convolution. Moreover, Pan et al. [30] proposed a multi-scale

channel attention module (MS-CAM) that relearns global and detail features from the features and integrates them together to reapply to the input feature map as attention weights, adjusting the semantics in global features and hidden detail information in local features simultaneously. They also proposed an attentional feature fusion (AFF) module that takes two feature maps of the same size as input. The module extracts global and local feature information through the MS-CAM module and effectively embeds and integrates the two feature maps using information from both aspects.

### 3. The Proposed Method

In this chapter, we provide a detailed description of the structure of our proposed model. In Section 3.1, we introduce the structure of our proposed model, which consists of two crucial components: the backbone network and the feature fusion network. Subsequently, in Section 3.2, we provide a detailed description of the backbone network structure, including the architecture and specific details of key modules within the network. Finally, in Section 3.3, we present an effective method for fusing semantic and detail features. This includes a detailed description of the fusion process, as well as the structure and implementation details of important modules used in the process.

### 3.1. Overall Architecture of DSDL-Net

DSDL-Net not only addresses issues of limited semantic understanding and difficult retention of detailed features in remote sensing object detection within complex background scenes but also proposes a novel approach to effectively fuse detailed features with high-level semantic features, enabling the network to fully leverage both for learning. Within the network, we propose two crucial components. The first is the adaptive high semantic learning and detail preservation (ASDP) network, which serves as the backbone. The second is an adaptive cross-level semantic–detail fusion (CSDF) network for feature fusion. The overall architecture of DSDL-Net is illustrated in Figure 1.

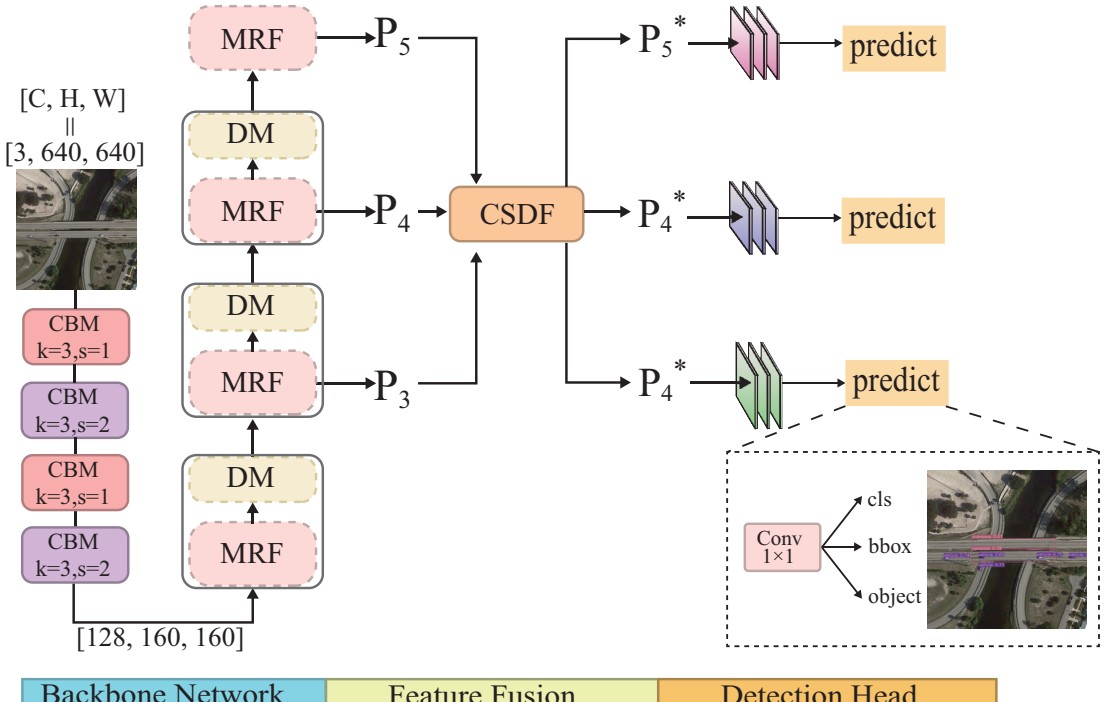

**Figure 1.** The overall structure of DSDL-Net.

The decoupling of semantics and fine-grained details collaboration learning in DSDL-Net is achieved through a two-step process. Firstly, the multi-receptive field feature fusion and detail mining (MRF-DM) module within the ASDP network is utilized to learn higher-

level semantic information during feature extraction while capturing rich, fine details and minimizing the loss of detail features. Secondly, the semantic–detail fusion (SDF) module in the CSDF network directly fuses these rich detail features with the feature map learned from the uppermost layer of the CSDF. As a result, the feature fusion network produces high-level semantic features that are rich in detail.

### 3.2. ASDP Network

In this section, we will introduce the structure of the backbone network ASDP in detail, including the important modules used and the details of various implementations of the network. The flowchart of the backbone network has been drawn in Figure 1.

The ASDP network accepts an input image of size 640 with 3 channels. The network begins with a stem block that reduces the image size by half twice. This finally produces feature maps $\{p1\}$ with spatial dimensions at 1/4 of the original image and channel counts of 128. After the stem block, the backbone network stacks MRF-DM modules to produce feature maps $\{p_2, p_3, p_4, p_5\}$ with spatial dimensions $\{1/4, 1/8, 1/16, 1/32\}$ of the original image and channel counts of $\{128, 256, 512, 1024\}$, respectively.

### MRF-DM Module

In the field of remote sensing, target detection models often struggle to effectively distinguish between objects and the background in images when dealing with scenarios with complex backgrounds. This is due to the insufficient semantic hierarchy learned by the model, resulting in false positives or missed detections. Additionally, the remote sensing images present a challenge for general backbone networks, as they often lose a significant amount of detail information during the feature extraction process, particularly for high-resolution images. To address these issues, we propose a novel multi-receptive field feature fusion and detail mining (MRF-DM) module, the overall structure of which is illustrated in Figure 2.

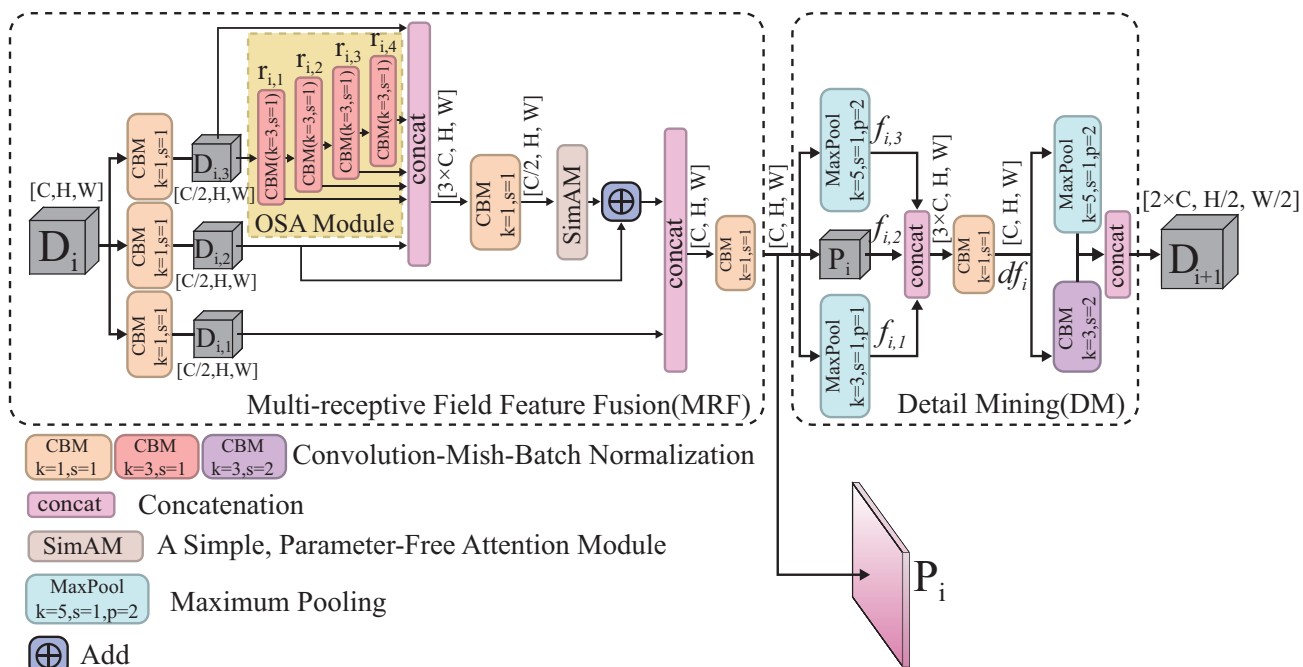

**Figure 2.** The overall structure of multi-receptive field feature fusion and detail mining (MRF-DM) module.

Firstly, the module employs a multi-receptive field feature fusion (MRF) structure. This structure utilizes an OSA [31] module to form multi-scale receptive field features $\{r_{i,1}, r_{i,2}, r_{i,3}, r_{i,4}\}$, achieving an equivalent effect to convolutional kernels of sizes $\{3, 5, 7, 9\}$.

The architecture of the OSA module is shown in Figure 3. By fusing multi-scale receptive field features, the MRF structure is able to obtain more contextual information and produce higher-level semantic information. The formula for calculating the generation of multi-scale receptive field features is presented in the following equation. The CBM convolutional layer consists of convolution, batch normalization, and a Mish activation function.

$$r_{i,j} = \begin{cases} CBM_{k=3,s=1}(D_{i,j}), & \text{if } j = 1 \\ CBM_{k=3,s=1}(r_{i,j-1}), & \text{if } j = 2,3,4 \end{cases} \tag{1}$$

In addition, our approach differs from the original OSA module in several ways. Firstly, we expand the structure's branches $\{D_{i,1}, D_{i,2}, D_{i,3}\}$ for incorporating low-level feature information into all feature fusion processes within the structure. This provides additional detail from low-level features, for example, $\{D_{i,2}, D_{i,3}\}$ are added to the fusion process of $\{r_{i,1}, r_{i,2}, r_{i,3}, r_{i,4}\}$, and $\{D_{i,3}\}$ is added to the final concatenate. Secondly, we design a residual connection between the high-level semantic output feature obtained by SimAM [28] and the initial branch. This serves to further emphasize the low-level features of the input data and ensure that the model can more accurately capture its details. Considering that SimAM's output is a multi-scale receptive field perspective of $\{D_{i,3}\}$, we choose branch $\{D_{i,2}\}$ as the object of residual connection to avoid feature redundancy. Finally, we employ the SimAM attention mechanism. As low-level detail features are mixed with high-level semantics, this attention mechanism can generate 3D weights for individual features without the need for additional parameters. This enables effective integration of semantic and detail features within the feature layer. The MRF structure is calculated as follows:

$$R_i = \text{concat}([r_{i,1}, r_{i,2}, r_{i,3}, r_{i,4}, D_{i,2}, D_{i,3}]) \tag{2}$$

$$S_i = \text{SimAM}(CBM_{k=1,s=1}(R_i)) + D_{i,2} \tag{3}$$

$$D_{i+1} = \text{CBM}_{k=1,s=1}(\text{concat}([S_i, D_{i,1}])) \tag{4}$$

The MRF structure has three advantages over the original OSA module: (1) the process of semantic learning and the enhancement of features using attention mechanisms both take into account the retention of as much low-level detail information as possible; (2) through the SimAM attention mechanism, all features are globally adjusted, effectively tuning and enhancing detail and semantic information; and (3) it does not bring many additional parameters, where the attention mechanism is parameter-free, and only two additional convolutions with a kernel size of 1 are added.

In conventional backbone networks, a learnable convolutional operation is typically used for pooling. However, traditional pooling processes often result in a loss of detailed features to some extent as the resolution of the feature maps decreases. To address this issue, we propose the simple structure of detail mining (DM) with reference to the SPP [32] module, which utilizes max pooling to generate three features $\{f_{i,1}, f_{i,2}, f_{i,3}\}$ that have the same size as the original feature map. These features are concatenated and compressed using a $1 \times 1$ convolutional kernel to generate the feature map $\{df_i\}$. By doing so, the network can capture detailed texture features from multiple scale features before pooling, reduce the loss of details during the pooling process, and compress the captured features to reduce the parameters. The specific equations for this process are shown below.

$$df_i = CBM_{k=1,s=1}\left(\text{concat}\left(\left[D_i, MaxPool_{k=3,p=1}(D_i), MaxPool_{k=5,p=2}(D_i)\right]\right)\right) \tag{5}$$

Subsequently, we employ a dual-path pooling approach using max pooling and adaptive convolution with a kernel size of 3 and a stride of 2 to downsample the feature map $\{df_i\}$, reducing its size by half. This downsampling operation retains a more comprehensive range of detailed features, thereby significantly reducing the loss of detail information

caused by decreased resolution during the pooling process. The computation formula for generating these features is as follows:

$$D_{i+1} = \text{concat}([CBM_{k=3,s=2}(df_i), MaxPool_{k=2,s=2}(df_i)])\qquad(6)$$

In summary, the ASDP network is used as a backbone network for object detection, specifically for feature extraction. Unlike traditional feature extraction networks, this network uses stacked OSA modules to extract and combine feature maps with varying receptive field sizes, enhancing the semantic levels of the features within the network. To preserve detailed features, we approach this from two angles. Firstly, we enhance the OSA module, resulting in the MRF structure. In this structure, low-level detailed features are integrated during each instance of multi-scale receptive field feature fusion, highlighting the importance of detailed features within the original feature map as part of the semantic learning process. Secondly, to prevent the loss of high-resolution detailed features due to feature pooling, we employ a DM structure for multi-scale pooling of detailed features. This mitigates the insufficiency of detailed features caused by single pooling. Then, a convolution with a kernel size of 1 is used to decrease the number of channels, compressing the captured detailed features and reducing the number of parameters. Thus, by using the MRF-DM structure, the network can enhance the semantic levels of features while preserving more detailed features during the learning process of the backbone network, and compress detailed features to reduce network parameters.

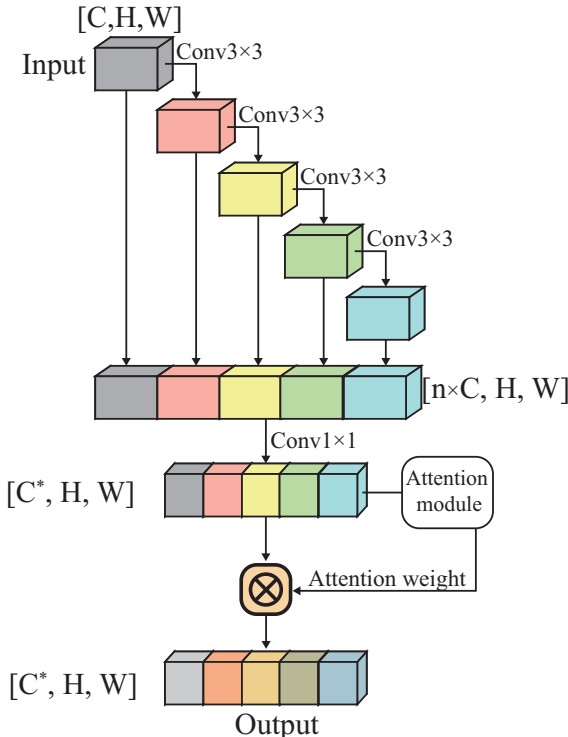

**Figure 3.** The overall structure of one-shot aggregation (OSA) convolutional.

### 3.3. CSDF Network

Many existing methods of integrating semantic and detailed features in learning may lead to overlearning, which can damage the quality of detailed features, and may also neglect the importance of maintaining high-quality semantic features during fusion. To address the above challenges, we propose the adaptive cross-level semantic–detail fusion (CSDF) network with reference to the PAN of Liu et al. [18]. The overall network structure is shown in Figure 4.

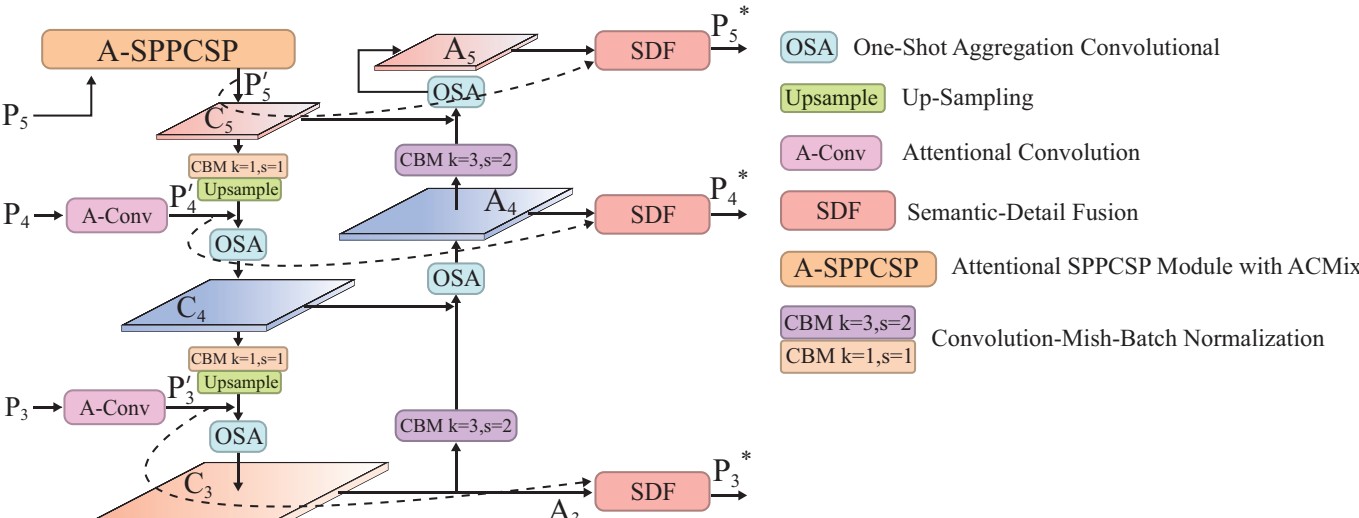

**Figure 4.** The structure of the adaptive cross-level semantic–detail fusion (CSDF) module includes the A-SPPCSP, A-Conv, and SDF modules, which are described in Sections 3.3.1, 3.3.2, and 3.3.3, respectively.

In the CSDF feature fusion network, to enhance and capture the detailed features in multi-scale feature maps, the output feature layer $\{P_5\}$ of the backbone network is first passed through the attentional SPPCSP (A-SPPCSP) module. Meanwhile, the feature maps $\{P_3, P_4\}$ are effectively enhanced with both global semantic and local detail texture information through the use of a lightweight attentional convolutional layer (A-Conv). As a result, the network obtains enhanced feature maps $\{P_3', P_4', P_5'\}$. Subsequently, the network employs two multi-scale feature fusion paths, top-to-bottom and bottom-to-top, to enrich contextual information. In the top–down path, the high-level features are upsampled and concatenated with the low-level feature maps enhanced by the A-Conv module from the left side. The concatenated features are then further learned by the OSA module. In the bottom–up pathway, high-level features are subjected to adaptive learnable pooling operations via convolution. Subsequently, the pooled feature maps are concatenated with low-level feature maps originating from the left side. Ultimately, the OSA module is employed once more to further learn these concatenated features. By doing so, the CSDF network obtains the highest semantic feature maps $\{A_3, A_4, A_5\}$. Subsequently, to effectively learn both the high-level semantic features and the detailed information present within the image, the CSDF network directly takes in the enhanced detailed feature maps $\{P_3', P_4' \, P_5'\}$ from the backbone network and adaptively fuses them with the highest semantic feature maps $\{A_3, A_4, A_5\}$ through the employment of the SDF module. The fusion module ensures that the rich detail features are embedded without incurring any loss of fine details due to excessive semantic learning. Additionally, before and after achieving joint learning of both semantic and detail features, this module ensures the consistency of semantics, preserves the original high-quality semantic features, and results in the generation of the final prediction feature layer $\{P_3^*, P_4^*, P_5^*\}$.

In our proposed CSDF network, we have incorporated two feature enhancement structures that have been proven to be effective. In the following Sections 3.3.1 and 3.3.2, we will introduce these structures in detail. We have redesigned and optimized these structures to better integrate them into our model. While these two structures may not be overly complex or represent groundbreaking innovation, they effectively enhance the key high-level semantic and fine-grained detail features within the network through a straightforward approach. Finally, in Section 3.3.3, we will discuss our approach to decoupling the fusion of semantic–detail information. In addition, a detailed introduction to the semantic–detail fusion (SDF) module used in the fusion process, as well as an overview of the module's process and implementation details, will be provided.

### 3.3.1. Attentional SPPCSP Module

The ACMix [29] module can effectively enhance the detail information in feature maps because it combines the strengths of both convolution and self-attention mechanisms. Convolution is effective in capturing local spatial relationships and patterns within the feature maps, while self-attention is effective in capturing long-range dependencies and relationships between different parts of the feature maps. By integrating these two mechanisms, ACMix can effectively capture both local and non-local information within the feature maps, resulting in a more comprehensive and detailed representation of the features.

For example, He et al. [33] introduced the use of ACMix to refine the complex background features in UAV aerial images. By incorporating ACMix after the backbone network, their experiments demonstrated that ACMix can effectively enhance and refine important features in complex background scenes by capturing both local spatial relationships and long-range dependencies within the feature maps. Similarly, Xue et al. [34] inserted ACMix after the backbone network and conducted object detection experiments on datasets containing very small objects. The results showed that ACMix can effectively help the network focus on small objects by capturing both local and non-local information within the feature maps, improving its ability to detect them. However, they also pointed out that ACMix has a large number of parameters, so in order to maintain the real-time performance of the model, ACMix cannot be added to the model in large quantities.

To enhance the representation of feature maps, we incorporated the ACMix attention module after the backbone network. We referred to the state-of-the-art SPPCSP [5] module and inserted the ACMix module after the first convolutional layer in the SPPCSP module, which reduces the number of input feature map channels by half. This reduction in input channels effectively reduces the number of parameters required for ACMix inference, allowing us to maintain real-time performance of the model while introducing a self-attention mechanism to the SPPCSP module. The overall structure of the attentional SPPCSP is illustrated in Figure 5.

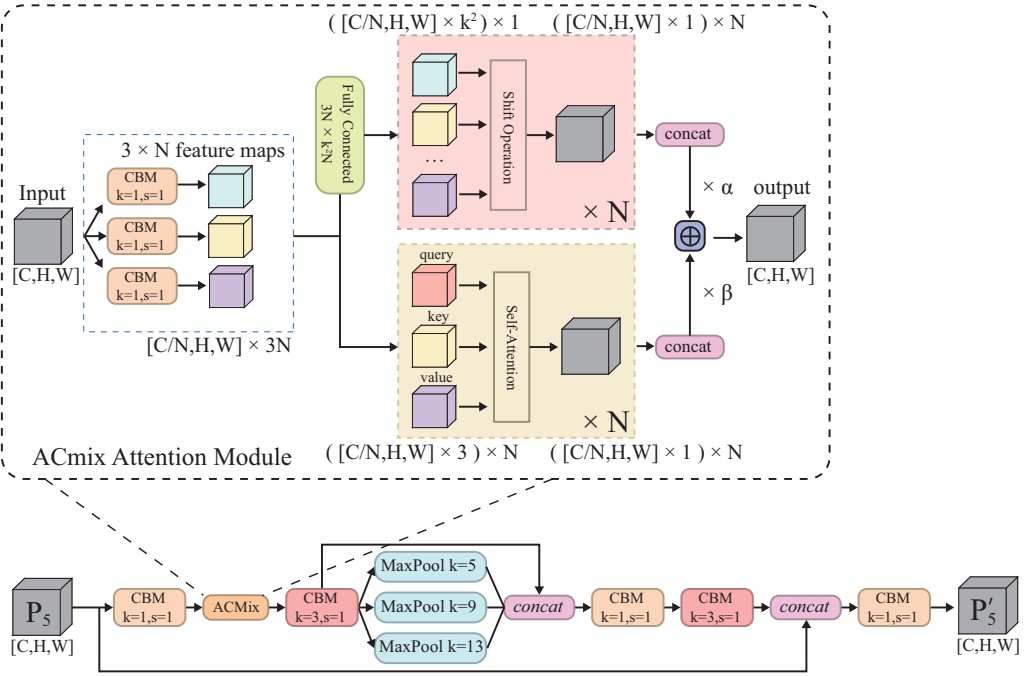

**Figure 5.** The entire structure of the attentional SPPCSP (A-SPPCSP) module.

The A-SPPCSP module takes an input $\{P_5\}$ and first applies a convolutional layer with a $1 \times 1$ kernel size to reduce the number of input channels by half. Then, the ACMix attention mechanism is utilized to focus on salient features. Subsequently, spatial pyramid pooling (SPP) is applied, which consists of a set of max-pooling layers with varying kernel

sizes, resulting in multi-scale max-pooled features. These features are then concatenated with the output of the ACMix layer and passed through a convolutional layer with a $1 \times 1$ kernel size to generate an enhanced final feature representation. The equation for the exact calculation of the above process is as follows.

$$t_1 = CBM_{k=3,s=1}(ACMix(CBM_{k=1,s=1}(P_5))) \tag{7}$$

$$t_2 = CBM_{k=1,s=1}(concat([t_1, MP_{k=5}(t_1), MP_{k=9}(t_1), MP_{k=13}(t_1)])) \tag{8}$$

Finally, the high-level features are passed through a convolutional layer with a $3 \times 3$ kernel size and then concatenated with the output of the first convolutional layer applied on the input tensor $\{P_5\}$, and then passed through a final convolutional layer to generate the output tensor $P_5'$. The final formula for calculating the flow of the network module is as follows.

$$P_5' = CBM_{k=3,s=2}(concat([CBM_{k=3,s=1}(t_2), P_5])) \tag{9}$$

### 3.3.2. Attentional Convolutional Layers

First of all, A-Conv is not a new module but rather a lightweight structure designed to enhance the feature layers $\{P_3, P_4\}$. It is simple to understand, yet highly effective. The reason for its effectiveness lies in the fact that the semantic features in $\{P_3, P_4\}$ require further learning within the CSDF network to enhance their semantics. Additionally, these layers also serve as rich sources of detail features that are transmitted to the SDF module for the final fusion of semantic and detail features. Our approach efficiently enhances both global and local features through lightweight learning. The overall structure of the A-Conv module is shown in Figure 6.

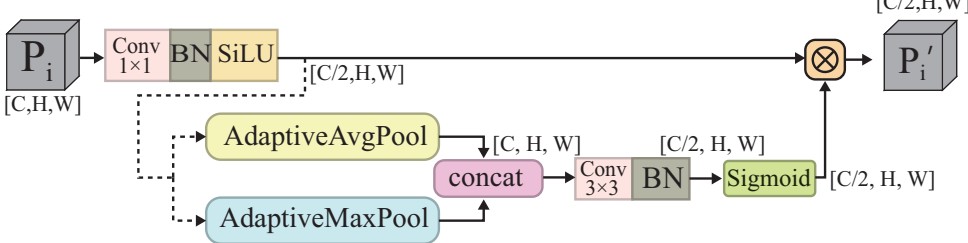

**Figure 6.** The overall structure of attentional convolutional layers (E-Conv).

We refer to Shen et al. [35], who proposed a multi-scale convolution attention module. This module uses averaging and maximum pooling to generate attention scores and performs these steps twice in series to further enhance important global and local features within the feature map. Finally, they also used the parameter-free 3D attention mechanism SimAM, introduced in the MRF-DM module, to simultaneously harmonize global semantics and layout detail information. Their experiments have demonstrated that this approach to forming attention can effectively balance global and local information within remote sensing datasets. By adjusting resource allocation based on the importance of the target, the neural network can focus more on important pixel areas while ignoring irrelevant regions.

In order to maintain the real-time and lightweight nature of the model, we simplified the steps of their module. Because we have already incorporated the SimAM attention mechanism into the MRF-DM module to control the coexistence between semantics and details, we designed a lightweight process that uses averaging and maximum pooling in a single inference step to generate attention weights for feature enhancement. This design effectively connects the features from the backbone network. This lightweight approach enhances important pixel areas within feature layers $\{P_3, P_4\}$, allowing fine details such as texture to be preserved for subsequent fusion with high-level semantic information. At the same time, feature maps $\{P_3, P_4\}$ can maintain a high level of semantics for transmission to CSDF for further learning at the semantic level.

The module first generates features $\{E_1\}$ by halving the number of channels through a convolutional layer. Then, the global and local contextual features $\{E_2\}$ are generated using adaptive averaging pooling and adaptive maximum pooling, respectively. Subsequently, the concatenation of the two features is passed through a convolutional layer to reduce the number of channels by half. Finally, the Sigmoid function is applied to the feature with half the number of channels just obtained to yield the attention weights $\{E_3\}$. These weights are then applied to $\{E_1\}$, resulting in the final output of the A-Conv module. The A-Conv feature enhancement module is calculated as follows.

$$E_1 = CBS_{k=1,s=1}(P_i) \tag{10}$$

$$E_2 = concat([AdaptiveAvgPool(E_1), AdaptiveMaxPool(E_1)]) \tag{11}$$

$$E_3 = Sigmoid_{k=1,s=1}(CB_{k=3,s=1}(E_2)) \tag{12}$$

$$P'_i = E_1 \cdot E_3 \tag{13}$$

The CBM convolutional layer, used in Section 3.3.1, consists of convolution, batch normalization, and a Mish activation function. In CBS, the activation function is replaced with SiLU, while in CB the Mish activation function is removed from the CBM convolutional layer.

### 3.3.3. Semantic–Detail Fusion (SDF) Module

The SDF module is designed to effectively fuse high-level semantic information with detailed information and represents the final and most crucial step in achieving semantic–detail learning. Figure 4 illustrates how the SDF module is utilized within the CSDF network to fuse the feature layers $\{P'_3, P'_4, P'_5\}$, generated from the features $\{P_3, P_4, P_5\}$ in the backbone network after the feature enhancement module, with the highest semantic-level feature layers $\{A_3, A_4, A_5\}$. The overall architecture of the SDF module is shown in Figure 7, where the calculation formula of the local contextual (LC) and global contextual (GC) feature extraction process is as follows. Where CBR refers to the combination of convolution, batch normalization, and ReLU activation function, AAP stands for adaptive average pooling, which downsamples the feature map to a size of $1 \times 1$.

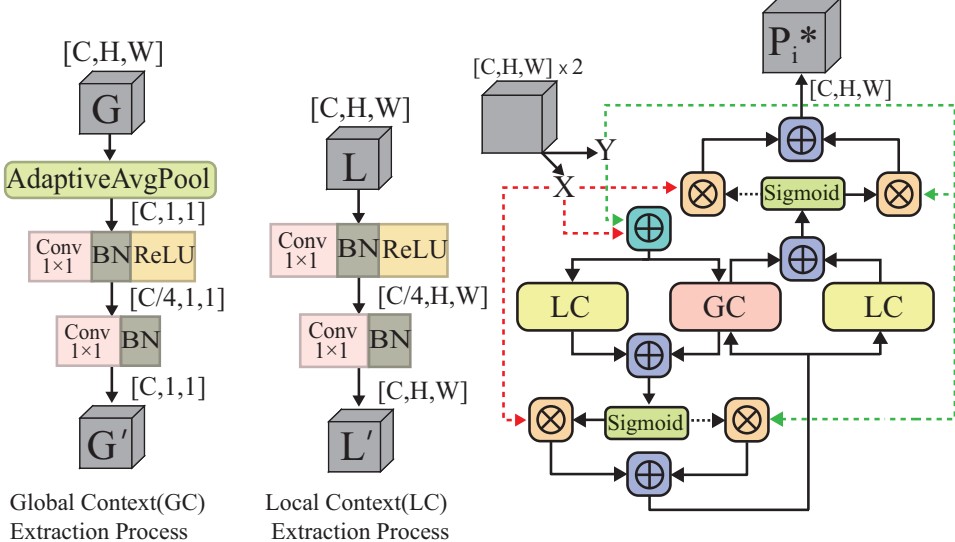

**Figure 7.** The overall structure of semantic–detail fusion (SDF) module.

$$G' = GC(G) = CB_{k=1,s=1}(CBR_{k=1,s=1}(AAP(G))) \tag{14}$$

$$L' = LC(L) = CB_{k=1,s=1}(CBR_{k=1,s=1}(L)) \tag{15}$$

We refer to the iAFF [30] module, which performs two parallel rounds of global–local context feature extraction (*GC* and *LC*). However, in our improved design, the learning of global context information is shared between the two rounds. This approach maintains the universality of global information and mitigates semantic inconsistency before and after fusion, ultimately achieving the mutual embedding of semantic and detail information. After fusion, with the addition of detailed information, the network can effectively utilize the extra detailed information to enhance its discriminative ability in local regions. Our subsequent experiments provide ample evidence of the effectiveness and robustness of our approach, successfully enhancing the network's overall capability to semantically recognize objects and accurately locate details within large-scale remote sensing images with complex backgrounds.

The SDF module takes in two input parameters, *X* and *Y*, adds them together, and passes the result to the global–local context feature extraction (*GC* and $LC_1$) module for learning global and local features. After the first round of learning, the outputs from the *GC* and $LC_1$ structures are added and passed through a Sigmoid function to generate an attention weight, $\{x_1\}$. This weight is then applied to *X* and *Y* in the form of $\{x_1, 1 - x_1\}$ and the result is added together to produce the first learning output, $\{x_2\}$. The second round of learning follows a similar process, with $\{x_2\}$ being passed to the global–local context feature extraction (*GC* and $LC_2$) module. Notably, the global context learning module used in this round is identical to that used in the first round, allowing for shared global information between both rounds of learning. Ultimately, an attention score of $\{x_3\}$ and a final output of $P_i^*$ are generated by the SDF module.

$$x_1 = Sigmoid(GC(X + Y) + LC_1(X + Y)) \tag{16}$$

$$x_2 = x_1 \cdot X + (1 - x_1) \cdot Y \tag{17}$$

$$x_3 = Sigmoid(GC(x_2) + LC_2(x_2)) \tag{18}$$

$$P_i^* = x_3 \cdot X + (1 - x_3) \cdot Y \tag{19}$$

The following describes how the final features $\{P_3^*, P_4^*, P_5^*\}$ used for object detection are generated by the SDF module. The network applies the feature enhancement module to generate features $\{P_3', P_4', P_5'\}$ from the initial features $\{P_3, P_4, P_5\}$. These features are then fused with the highest semantic-level features $\{A_3, A_4, A_5\}$ using the SDF module to obtain the final features $\{P_3^*, P_4^*, P_5^*\}$. Feature maps $\{C_3, C_4, C_5\}$ are generated through a top–down path, while features $\{A_3, A_4, A_5\}$ are produced via a bottom–up feature fusion path. The formula for this process is as follows:

$$P_i^* = \text{SDF}(P_i', A_i) \quad i = 3, 4, 5 \tag{20}$$

Overall, we use an appropriate and improved method with proven effectiveness for attention-enhanced features in the CSDF module. Then, we use a novel approach to fuse semantic and detail features. The overall contribution can be divided into the following two points:

1. We adapted the iAFF module to fuse detail and semantic information. Unlike the original, we did not treat both equally during fusion. Instead, we proposed a method that learns local details twice with global semantic sharing. This embeds detail without compromising learned semantics, effectively fusing both types of information.
2. We deviated from the common practice of fusing various features within the pyramid structure. Instead, we directly obtained feature maps rich in detail information from the backbone network and fused them with the highest semantic-level feature

layers $\{P_3^*, P_4^*, P_5^*\}$. This approach reduces the loss of detail information and further diminishes the coupling relationship between details and semantics.

## 4. Experiments

In this section, we first introduce the datasets used in our experiments and the metrics used for validation. We conducted extensive ablation experiments on multiple datasets to demonstrate the robustness and effectiveness of our model. In addition, we conducted comparative experiments, comparing our proposed DSDL-Net with various popular baseline models in the field. Finally, we analyzed our experimental results in detail, including a visual analysis of our model's detection results and an analysis of detection accuracy for specific categories in complex background remote sensing images.

### 4.1. The Dataset and Evaluation Metrics

#### 4.1.1. Experimental Datasets

To thoroughly demonstrate the strength of our model and the success of our approach, we conducted ablation experiments on three publicly available remote sensing datasets: DIOR [36], TGRS-HRRSD [37], and RSOD [38]. Specifically, in order to compare our proposed DSDL-Net with other models in a fair manner, we chose the largest DIOR remote sensing dataset for the comparative experiment.

The DIOR dataset is a large-scale optical remote sensing image object detection dataset and is currently the largest in its field. It contains 23,463 images with 192,472 instances across 20 object categories. The dataset is already divided into training (5862 images), validation (5863 images), and testing (11,738 images) sets. We conducted all our experiments using this official division.

The TGRS-HRRSD dataset, designed for high-resolution remote sensing image object detection, contains 21,761 images across 13 categories. It has been pre-divided into training (5401 images), validation (5417 images), and testing (10,943 images) sets. We used this official division for all our experiments.

The RSOD dataset is an open remote sensing image object detection dataset with four categories: airplanes (446 images with 4993 instances), oil tanks (165 images with 1586 instances), playgrounds (189 images with 191 instances), and overpasses (176 images with 180 instances). The annotations are in the PASCAL VOC format. Because there is no official division, we divided the dataset into training (747 images), validation (94 images), and testing (95 images) sets in an 8:1:1 ratio. We divided each category of images randomly according to this ratio to ensure that all sets contained a proportional representation of each category.

#### 4.1.2. Ablation Experiment Metrics

In the ablation experiment, we refer to the Pascal VOC2012 standard to calculate average precision ($AP$), and its calculation formula is as follows.

$$AP = \int_0^1 p(r)dr \qquad (21)$$

In object detection, average precision ($AP$) is calculated by integrating the area under the precision–recall curve. Here, $p(r)$ represents the precision value at a recall rate of $r$. The range of integration is from 0 to 1, covering all possible values of recall. The precision–recall curve describes the relationship between precision and recall at different confidence thresholds. $AP$ is calculated by integrating the area under this curve, providing a comprehensive measure of the model's performance across different confidence thresholds.

In our ablation experiments, we used a variety of metrics to evaluate the performance of our model. These included mean average precision ($mAP$) for all classes at 0.50 IoU thresholds, precision, recall, $F1$-*score*, the number of parameters (Params), and GFLOPs. The formulas for calculating some of these evaluation metrics are shown below.

$$Precision = \frac{TP}{TP + FP} \tag{22}$$

$$Recall = \frac{TP}{TP + FN} \tag{23}$$

$$F_1\text{-}score = \frac{2 \cdot Precision \cdot Recall}{Precision + Recall} \tag{24}$$

$$mAP = \frac{1}{N} \sum_{1}^{N} AP_i \tag{25}$$

When computing $mAP$, the variable $N$ represents the number of categories detected by the model and $AP_i$ denotes the average precision of the $i$-th category. For precision and recall calculations, $TP$ refers to instances where the model correctly predicts a positive class for a given sample. In contrast, $FN$ indicates cases where the model incorrectly predicts a negative class for a positive sample. Finally, $FP$ represents instances where the model incorrectly predicts a positive class for a negative sample.

### 4.1.3. Comparative Experiment Metrics

In our comparative experiment, we followed the standards of the MS COCO dataset and used the official Pycocotools tool to evaluate the performance of our model on the test set. We utilized several evaluation metrics calculated by the tool, including average precision at 0.50 IoU threshold ($AP_{50}$), $AP_{small}$ for instances with an area smaller than $32^2$ pixels, $AP_{medium}$ for instances with an area between $32^2$ and $96^2$ pixels, and $AP_{large}$ for instances with an area larger than $96^2$ pixels. These metrics fully assess the performance of different models in detecting instances of different scales and achieving a 0.50 IoU threshold over all classes. Additionally, we also used the number of parameters (Params) and GFLOPs as evaluation metrics in our comparative experiment.

### 4.2. Implementation Details

For all ablation experiments, we conducted training and testing on a machine equipped with an Intel Core i5-12600KF CPU, 16GB of memory, and an NVIDIA TRX 3080 GPU with CUDA 11.7 support. Our models were implemented using PyTorch 1.13.1. All of our ablation experiments used a batch size of 8. We did not use pre-trained models or the warmup training strategy in our experiments. During the initial training phase, we initialized the learning rate to $1 \times 10^{-2}$ and used the cosine annealing learning rate for training. The learning rate was decayed every 1000 steps, with a final learning rate of 0.1 times the initial learning rate and an IoU training threshold of 0.2. We applied data augmentation techniques such as random left–right flipping with a probability of 0.5, mosaic data augmentation with a probability of 1, and mix-up data augmentation with a probability of 0.15. The input image size was set to $640 \times 640$ for the DIOR and TGRS-HRRSD datasets, and these models were trained for 500 epochs. For the RSOD dataset, the input image size was set to $416 \times 416$ and the model was trained for 600 epochs.

Most of the comparative experiments were conducted on a machine equipped with an NVIDIA TRX 3090 GPU, CUDA 11.6, mmcv 1.6.1 and mmdetection 2.25.1, using the DIOR dataset for training and testing. The experiments were trained for 500 epochs, with a batch size of 8 and an initial learning rate of $1 \times 10^{-2}$. The remaining configurations used the default settings of the baseline models in the framework. Some experiments employed mosaic and mix-up data augmentation techniques, which are indicated by an asterisk (*) in the table.

In our loss function, we used a weighted target detection loss that includes the localization loss ($L_{box}$), object loss ($L_{obj}$), and classification loss ($L_{cls}$). For the localization loss, we used the well-known $CIoU$ to evaluate the positional relationship between the predicted and ground-truth bounding boxes. While $IoU$ is commonly used to measure the error of bounding box regression, it suffers from slow convergence and inaccurate regres-

sion when detecting small objects. To address these issues, Zheng et al. [39] introduced the *CIoU* loss, which takes into account the overlap size of bounding boxes, the distance between their center points, and their aspect ratio. The definition of *IoU* is shown in the following equation:

$$IoU = \frac{A \cap B}{A \cup B} \tag{26}$$

where *A* and *B* are the ground-truth box and predicted box, respectively. The penalty term can be represented as

$$v = \frac{4}{\pi^2} \left( \arctan \frac{w^{gt}}{h^{gt}} - \arctan \frac{w}{h} \right)^2 \tag{27}$$

where $w^{g}t$ and $h^{g}t$ are the width and height of the ground-truth box, and *w* and *h* denote the width and height of the predicted box. $\alpha$ is a positive trade-off parameter, as seen in Equation (14):

$$\alpha = \frac{v}{(1 - IoU) + v} \tag{28}$$

The localization loss function can be defined as

$$L_{box} = 1 - IoU + \frac{\rho^2(b, b^{gt})}{c^2} + \alpha v \tag{29}$$

In this center point-based approach to object detection, we use the symbols *b* and $b^{g}t$ to represent the center points of the predicted and ground-truth frames, respectively. The Euclidean distance between these two center points is denoted by $\rho^2$, while *c* refers to the diagonal length of the smallest closed area that contains both frames.

To calculate the loss associated with confidence and classification, we use the cross-entropy loss function as a standard measure of the mismatch between the predicted output and true labels. The formula for this loss function is as follows:

$$BCE_{Loss} = L_{obj} = L_{cls} = -(y \cdot \log(\hat{y}) + (1 - y) \cdot \log(1 - \hat{y})) \tag{30}$$

where *y* denotes the ground-truth label, $\hat{y}$ denotes the predicted label, and $BCE_{Loss}$ denotes the loss between the model output and the ground-truth label. The overall loss function is represented by the following equation:

$$\text{Loss} = W_1 \cdot L_{box} + W_2 \cdot L_{cls} + W_3 \cdot L_{obj} \tag{31}$$

In this paper, we assign weights to three types of losses, namely, the localization loss $W_1$, classification loss $W_2$, and object confidence loss $W_3$, denoted as {0.05,0.3,0.7}, respectively.

### 4.3. Ablation Experiments

In this section, we perform a comparative analysis of our experimental results from the ablation experiments we conducted. We conducted ablation experiments on three datasets, each consisting of four experiments: a baseline model, followed by two performance testing experiments that, respectively, added the MRF-DM network and the CSDF network, in order to compare the effects of the two methods alone compared to the baseline. Finally, we tested the performance of our proposed model by combining both networks with the baseline, which allowed us to visually compare the results of this experiment with the previous three.

Tables 1–3 show the results of our ablation experiments, where we evaluated the MRF-DM and CSDF networks with YOLOv4-CSP as a baseline model on three remote sensing datasets. Taking Table 1 as an example, applying the MRF-DM module alone reduced the model's parameters and GFLOPs by 4.2 M and 4.82 G, respectively, while improving the *mAP@0.5*, precision, recall, and *f*1-score by 1.63%, 1.86%, 0.42%, and 0.96%,

respectively. Similarly, the CSDF network alone reduced the model's parameters and GFLOPs by 3.2 M and 5.36 G, respectively, while improving the $mAP@0.5$, precision, recall, and $f1$-score by 1.58%, 1.38%, 0.55%, and 0.79%, respectively. Combining both networks further reduced the model's parameters and GFLOPs by 7.4 M and 10.18 G, respectively, while improving the $mAP@0.5$, precision, recall, and $f1$-score by 2.19%, 1.22%, 1.85%, and 1.50%, respectively. These results demonstrate the effectiveness of the proposed networks in enhancing the model's performance and stability. Compared to the baseline model, our model's $mAP@0.5$ reached 79.74%, representing a 2.19% improvement, with a parameter reduction of approximately 14%.

**Table 1.** Experimental Results of Each Module on the DIOR Dataset (VOC Format).

| Baseline | MRF-DM | CSDF | $mAP@0.5$(%) | Precision (%) | Recall (%) | F1 (%) | Params (M) | GFLOPs |
|---|---|---|---|---|---|---|---|---|
| ✓ | | | 77.55 | 84.85 | 72.51 | 77.84 | 52.57 | 119.21 |
| ✓ | ✓ | | 79.18 (+1.63) | **86.70 (+1.86)** | 72.94 (+0.42) | 78.80 (+0.96) | 48.36 (−4.20) | 114.39 (−4.82) |
| ✓ | | ✓ | 79.13 (+1.58) | 86.23 (+1.38) | 73.06 (+0.55) | 78.63 (+0.79) | 49.37 (−3.20) | 113.85 (−5.36) |
| ✓ | ✓ | ✓ | **79.74 (+2.19)** | 86.07 (+1.22) | **74.36 (+1.85)** | **79.34 (+1.50)** | 45.17 (−7.40) | 109.03 (−10.18) |

**Table 2.** Experimental Results of Each Module on the TGRS-HRRSD Dataset (VOC Format).

| Baseline | MRF-DM | CSDF | $mAP@0.5$(%) | Precision (%) | Recall (%) | F1 (%) | Params (M) | GFLOPs |
|---|---|---|---|---|---|---|---|---|
| ✓ | | | 92.70 | 90.05 | 88.95 | 89.36 | 52.53 | 119.09 |
| ✓ | ✓ | | 94.52 (+1.82) | **92.78 (+2.73)** | 91.17 (+2.22) | 91.89 (+2.52) | 48.32 (−4.21) | 114.27 (−4.82) |
| ✓ | | ✓ | 93.12 (+0.42) | 91.34 (+1.29) | 89.36 (+0.41) | 90.26 (+0.90) | 49.33 (−3.20) | 113.73 (−5.36) |
| ✓ | ✓ | ✓ | **94.66 (+1.96)** | 92.70 (+2.65) | **91.51 (+2.56)** | **92.03 (+2.67)** | 45.13 (−7.40) | 108.91 (−10.18) |

**Table 3.** Experimental Results of Each Module on the RSOD Dataset (VOC Format).

| Baseline | MRF-DM | CSDF | $mAP@0.5$(%) | Precision (%) | Recall (%) | F1 (%) | Params (M) | GFLOPs |
|---|---|---|---|---|---|---|---|---|
| ✓ | | | 92.60 | 93.10 | 87.28 | 90.06 | 52.48 | 118.93 |
| ✓ | ✓ | | 93.31 (+0.71) | **93.82 (+0.72)** | 91.87 (+4.59) | 92.73 (+2.67) | 48.28 (−4.20) | 114.11 (−4.82) |
| ✓ | | ✓ | 94.57 (+1.97) | 93.46 (+0.36) | 90.44 (+3.16) | 91.86 (+1.80) | 49.28 (−3.20) | 113.57 (−5.36) |
| ✓ | ✓ | ✓ | **95.56 (+2.96)** | 91.67 (−1.43) | **95.91 (+8.63)** | **93.69 (+3.63)** | 45.08 (−7.40) | 108.75 (−10.18) |

Overall, the results of the three ablation experiments indicate that both the MRF-DM and CSDF networks have a positive effect on improving the baseline performance in terms of the $mAP@0.5$, demonstrating the effectiveness of the two proposed methods. It can be observed from the data of the three experiments that using the MRF-DM network can significantly improve the precision of the network, proving that the MRF-DM backbone network can effectively solve the problem of false negatives caused by an insufficient semantic level in complex background object detection. Regarding the impact of the CSDF structure on the overall network, when used alone, it results in an increase in the $mAP@0.5$, precision, and recall compared to the baseline. However, its enhancement of network precision is not as pronounced as that of the MRF-DM module, which is capable of improving the network's semantic recognition ability. Nonetheless, when both the MRF-DM and CSDF modules are applied to the baseline, the model achieves an optimal $mAP@0.5$ and $F1$-*Score*, demonstrating their complementary nature. It should be noted, however, that the precision achieved when both modules are applied is still lower than when only the MRF-DM module is used. This suggests that the final fusion of detailed features in the network does have a somewhat detrimental effect on the originally perfect semantic features, reducing the network's semantic recognition ability and decreasing the model's detection precision. Despite this, the overall performance of the network remains optimal. Additionally, the experimental data also revealed that the combination of the two networks exceeded the effect of either network acting alone on the baseline network,

which indirectly verifies the correctness of our method. Firstly, our method effectively improves the semantic level that the backbone network can learn from through the MRF-DM network, while retaining rich and detailed features. Secondly, the CSDF network reasonably integrates the preserved fine details with the further learned semantic features, enabling the network to achieve joint learning of semantics and details.

### 4.4. Comparative Experiment

The DIOR remote sensing dataset, which is publicly available, was used in the comparative experiment. The data results for each experiment were obtained by testing the model on the test set, training it on the training set, and selecting the final fully converged model using the validation set. The test dataset contains a total of 11,738 images and 124,443 target instances. In addition to the comparative experiments, we conducted further experiments on this dataset, and in Section 4.5, we provide a visual analysis. We will introduce this dataset in greater detail below. As shown in Figure 8, we counted the number of instances for each category in the dataset and presented the results in a bar chart. From this chart, it can be seen that categories such as ship, storage tank, tennis court, and vehicle have a relatively large number of instances, while other categories generally have no more than 10,000 instances. Additionally, in Figure 9, we plotted the number of small, medium, and large object instances in the official division of training, validation, and test sets. In the bar charts for the "validation set" and "training set", the number of instances for all three types of objects is roughly equal. However, in the bar chart for the "test set", there are more instances of large and small objects and fewer instances of medium-sized objects.

We compare our proposed DSDL network with various baseline models of object detection, and we analyze the experimental results of the comparative experiments. The models involved in the comparative experiments are classified into three categories: anchor-free, one-stage, and two-stage models. The results of the comparative experiment are shown in Table 4.

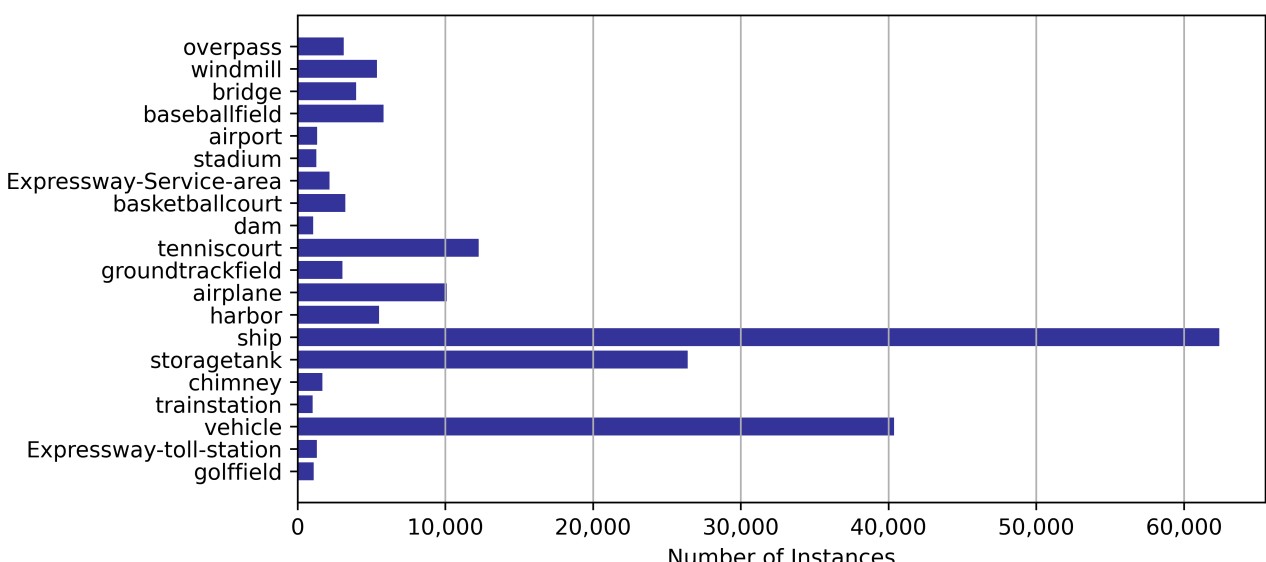

**Figure 8.** Number of instances for each category in the DIOR dataset.

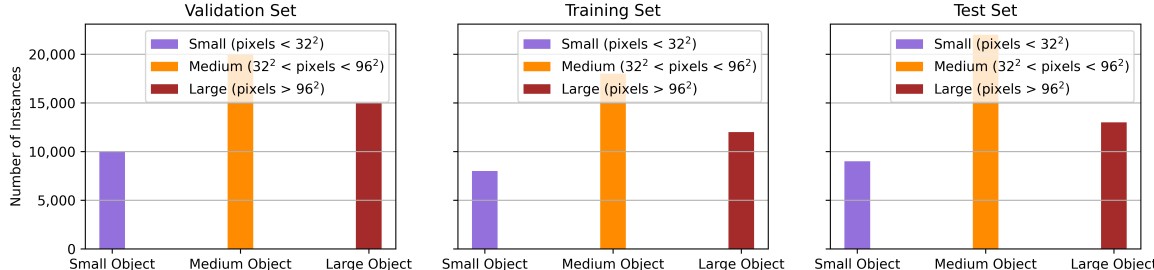

**Figure 9.** Number of small, medium, and large object instances in the DIOR dataset.

**Table 4.** Comparison of two-stage, anchor-free, and anchor-based detection algorithms on DIOR (COCO format). An asterisk (*) indicates the use of mosaic and mix-up data augmentation techniques in the experiments.

| Method | Backbone | $AP_{50}$ (%) | $AP_{small}$ (%) | $AP_{mdeium}$ (%) | $AP_{large}$ (%) | Params (M) | GFLOPs (G) |
|---|---|---|---|---|---|---|---|
| FSAF [40] | ResNet-101-FPN | 58.2 | 6.3 | 26.5 | 55.8 | 55.1 | 279.4 |
| ATSS [41] | ResNet-101-FPN | 59.8 | 5.9 | 28.1 | 58.6 | 50.9 | 278.4 |
| RepPoints [11] | ResNet-101-FPN | 63.2 | 5.8 | 29.7 | 61.7 | 55.6 | 266.0 |
| YOLOX-L * [42] | CSPDarknet-53 | 73.2 | 16.2 | 40.7 | 68.1 | 54.2 | 194.3 |
| Faster R-CNN [7] | ResNet-50-FPN | 59.1 | 5.7 | 27.0 | 56.0 | 41.2 | 206.8 |
| Cascade R-CNN [8] | ResNet-50-FPN | 59.5 | 6.0 | 28.8 | 60.7 | 69.0 | 234.5 |
| Gride R-CNN [9] | ResNet-50-FPN | 60.7 | 7.0 | 30.1 | 62.0 | 64.3 | 320.2 |
| Libra R-CNN [10] | ResNet-50-FPN | 59.5 | 5.9 | 28.1 | 56.9 | 41.5 | 207.8 |
| VarifocalNet [43] | ResNet-101-FPN | 61.3 | 7.3 | 30.2 | 61.8 | 51.5 | 266.0 |
| Guided-Anchoring [44] | ResNet-101-FPN | 64.4 | 7.5 | 30.9 | 59.7 | 56.1 | 273.5 |
| YOLOv4-CSP * [5] | CSPDarknet-53 | 76.6 | 16.1 | 42.8 | 72.1 | 52.6 | 119.2 |
| YOLOR-CSP * [6] | CSPDarknet-53 | 77.6 | 16.3 | 44.3 | 74.0 | 52.6 | 119.2 |
| DSDL-Net | ASDP | 66.7 | 11.6 | 33.8 | 62.6 | 45.2 | 109.0 |
| DSDL-Net * | ASDP | 78.7 | 17.5 | 44.8 | 75.3 | 45.2 | 109.0 |

In this experiment, by using mosaic and mix-up data augmentation techniques, our model outperformed some of the most popular models on the $AP_{50}$ metric. For example, in the anchor-free model, our model exceeded the $AP_{50}$ of YOLOX-L by 5.5%. In the one-stage model, it exceeded YOLOR-CSP by 1.1% and YOLOv4-CSP by 2.1%. Even without strong data augmentation, our model performed better than all the compared baseline models. For instance, in the two-stage model, our model improved the $AP_{50}$ metric by 7.6%, 7.2%, and 6% compared to Faster R-CNN, Cascade R-CNN, and Grid R-CNN, respectively. For the anchor-free models FSAF, ATSS, and RepPoints, it improved by 8.5%, 6.9%, and 3.5%, respectively. In the one-stage models, our model also outperformed Varifocal-Net and Guided-Anchoring, with improvements of 5.4% and 2.3%, respectively.

In general, based on the results of our comparative experiments, our proposed DSDL network model outperforms current mainstream YOLO series models such as YOLOv4-CSP, YOLOX, and YOLOR-CSP when using strong data augmentations like mix-up and mosaic. Furthermore, even under the same conditions without these augmentations, our model still surpasses all the compared baseline models, demonstrating the superior performance of our proposed model. Additionally, our model has fewer parameters and computational complexity than most of the models involved in the comparison, and our experimental results confirm the effectiveness of our proposed method in the domain of remote sensing object detection, especially in complex background scenarios.

### 4.5. Visualization of Detection Results

In this section, initially, in the object detection task using the DIOR dataset, we compute precision–recall (PR) curves for each category and calculate the mean average precision metric ($mAP@0.5$) with an IoU threshold of 0.5. Following this, we also plot $F1$-*Score* curves to show the changes in the $F1$-*Score* for each category at different confidence levels and the

confidence levels corresponding to the highest *F1-Score* for all categories. Finally, we also visualize and compare the detection performance of the same category detection algorithm in a real-world detection scenario.

### 4.5.1. Visualization of Precision–Recall (PR) and *F1-Score* Curve

As shown in Figures 10 and 11, our experimental results indicate a significant performance disparity among different categories. However, overall, our model exhibits strong performance in handling complex backgrounds. This can be attributed to the effective design of our model, which enables collaborative learning of image semantics and detailed information, allowing for accurate object detection in various complex scenarios.

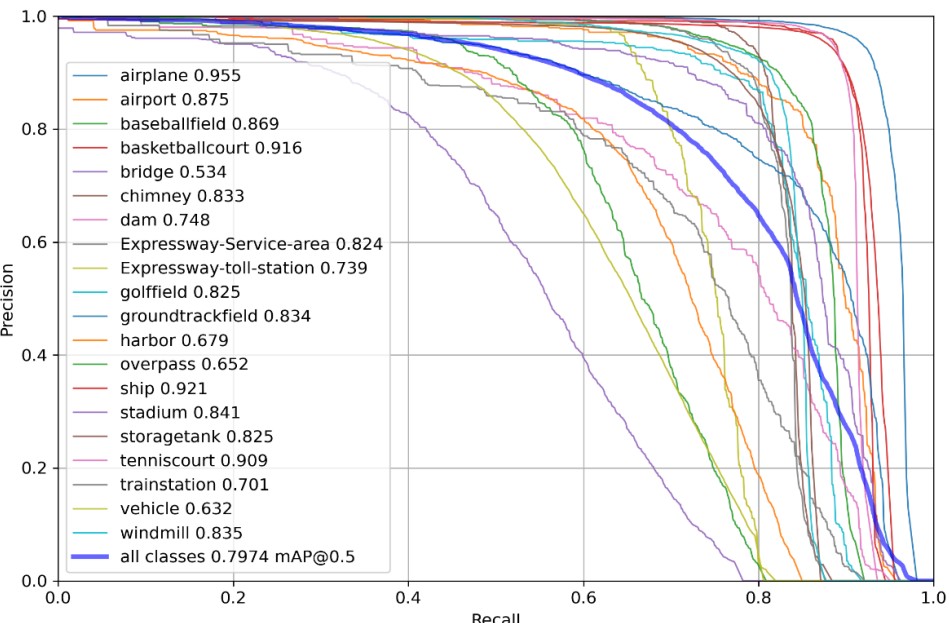

**Figure 10.** Precision–recall curves for individual categories on the DIOR dataset: different colors represent different categories, with the thickest blue curve representing the average precision–recall curve for all categories, and each category followed by its average precision (AP) value, which is the area under its precision–recall curve.

Through our experimentation, we discovered that the *F1-Score* of the model achieved its peak value of 0.7934 on the entire dataset when the confidence threshold was established at 0.2833. In addition, it is noteworthy that our model demonstrates high accuracy in detecting targets, such as airplanes ($mAP@0.5 = 0.955$), basketball courts ($mAP@0.5 = 0.916$), ships ($mAP@0.5 = 0.921$), and tennis courts ($mAP@0.5 = 0.909$). These categories are often present in complex remote sensing image backgrounds, such as urban buildings, water areas, and sports venues. The model's ability to successfully identify targets in these complex backgrounds signifies its robustness and generalization capabilities.

Nonetheless, there is still room for improvement in detection performance for certain categories, such as bridges, harbor, overpasses, railway stations, and vehicles. These performance gaps may be associated with the unique attributes of these categories in remote sensing images. For instance, overpasses ($mAP@0.5 = 0.652$) and bridges ($mAP@0.5 = 0.534$) in remote sensing images are generally characterized by multi-layer intersecting road structures, resulting in high spatial complexity. This may hinder the object detection algorithm from fully capturing the spatial structural information of overpasses and bridges, leading to reduced detection performance. Additionally, the railway station ($mAP@0.5 = 0.701$) category contains numerous linear features, such as tracks and platforms, which may be confused with other similar structures in remote sensing images, thus increasing the detection difficulty. For the harbor ($mAP@0.5 = 0.679$), although the scene is a simple back-

ground of sea surface remote sensing images, the high-density ships in the port cause some structures of the port to be obscured, disrupting the semantic recognition and positioning of the port. For vehicle detection ($mAP@0.5 = 0.632$), it is generally caused by extremely small targets. For high-resolution remote sensing images, extremely small target detection is very challenging for model robustness, which requires the model to meet the requirements of remote sensing target detection while being able to cope with small target detection. In view of the above problems, challenging remote sensing target detection categories usually have complex spatial objects, unusual shapes, high-density occlusion detection, small target detection, and other issues. Improving the detection capabilities of specific difficult-to-detect categories such as bridges, harbor, overpasses, railway stations, and vehicles will have a positive impact on traffic safety, maritime detection and management, transportation efficiency, and urban planning.

Despite these challenges, our model achieved an overall $mAP@0.5$ performance of 0.7974, proving its effectiveness in remote sensing image object detection tasks within complex backgrounds. To further substantiate this, we will conduct comparative experiments in subsequent sections.

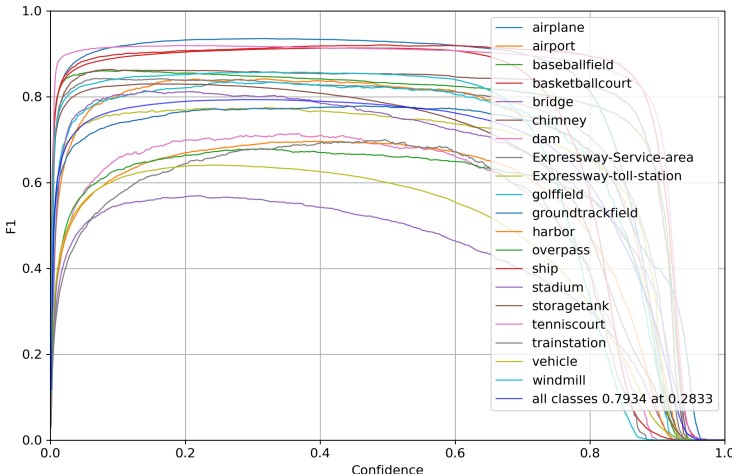

**Figure 11.** *F*1-*Score* curves for individual categories on the DIOR dataset.

### 4.5.2. Comparison of Detection Performance Visualization in Real Scenarios

As illustrated in Figure 12, our analysis of the images reveals that the DSDL network exhibits enhanced capabilities in terms of semantic recognition and localization. In the complex background scenario of urban remote sensing images, the DSDL network displays superior performance in semantic recognition and localization. For instance, in complex urban backgrounds, the DSDL network demonstrates robust semantic recognition capabilities for small and medium-sized objects, effectively reducing the occurrence of false positives.

In simple background scenarios, such as the seawater background of harbors and the uncluttered background of airports, our model exhibits strong robustness. Although the seawater background of the harbor is relatively simple, it presents challenges related to high-density object detection. For example, in a simple scenario with multiple harbors and a large number of ships, the DSDL network outperforms both the baseline model and the YOLOR-CSP model in terms of harbor recognition accuracy. Additionally, amidst the interference of ships, only our proposed network accurately locates the boundary position of the harbor. Similarly, in airport scenes, while all comparison models correctly detect all airplanes, only our model detects all vehicles. Vehicles appear in the image with very few pixels, further demonstrating the robustness of our model.

In summary, our model exhibits superior recognition and localization capabilities in complex backgrounds while maintaining strong robustness when facing simple backgrounds. It can effectively cope with more complex and changeable scenes and has excellent detection capabilities for multiple targets and small targets.

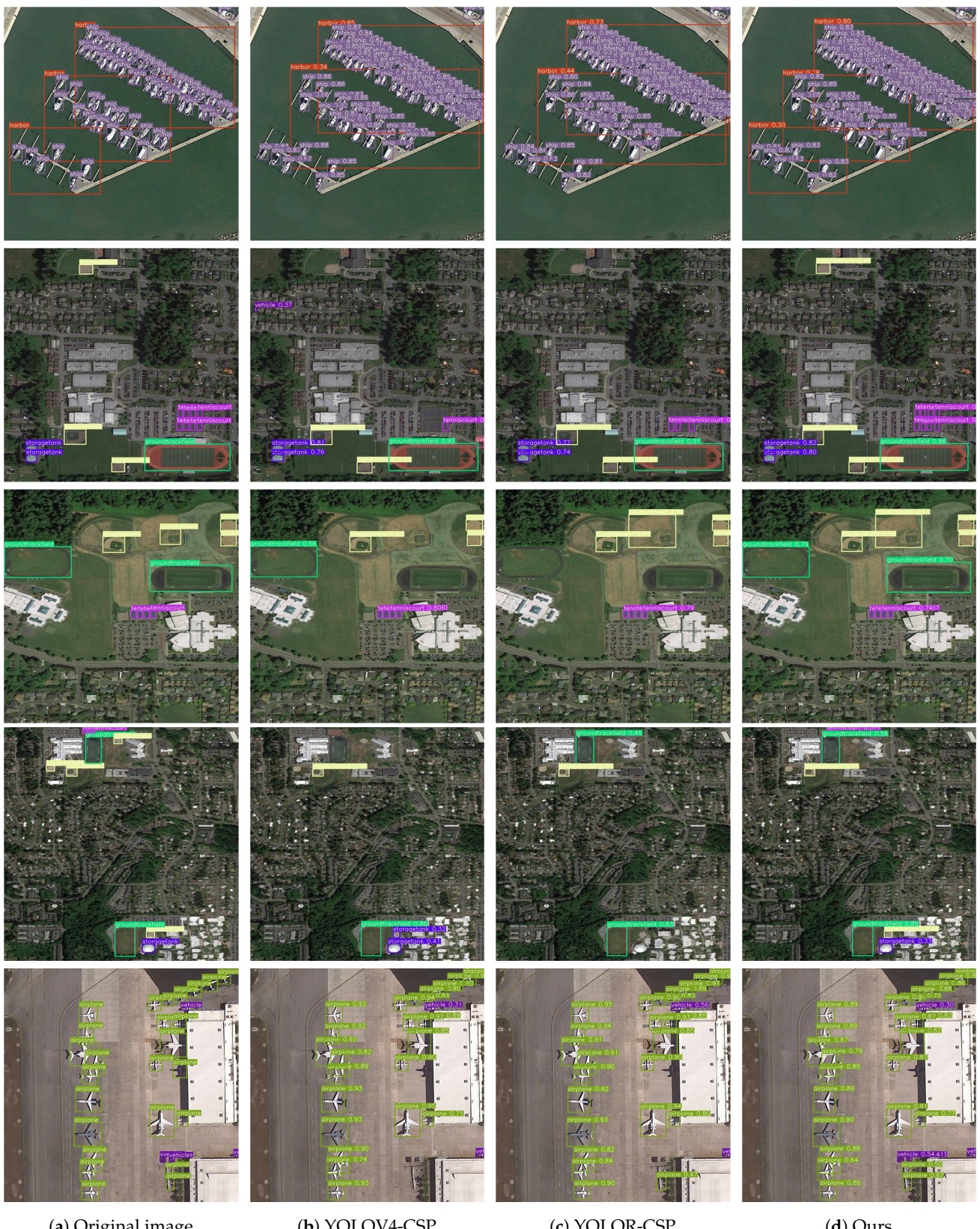

(**a**) Original image  (**b**) YOLOV4-CSP  (**c**) YOLOR-CSP  (**d**) Ours

**Figure 12.** Comparison diagram of the actual detection of different algorithms.

*4.6. Computational Requirements and Training Time of the DSDL-Net*

In this section, we present a detailed analysis of the training process of our model on the large-scale DIOR remote sensing dataset. We discuss the convergence behavior of

the network, the time required for training and validation per epoch, and the memory requirements during training.

Figure 13 shows the training log of our model, where the blue line represents the validation mean average precision ($mAP$) at an intersection over union (IoU) threshold of 0.5 for each epoch, and the purple line represents the validation $mAP$ at an IoU threshold ranging from 0.5 to 0.95. As can be seen from the figure, our model approaches convergence at around 350 epochs, after which there is no significant improvement or fluctuation in its performance.

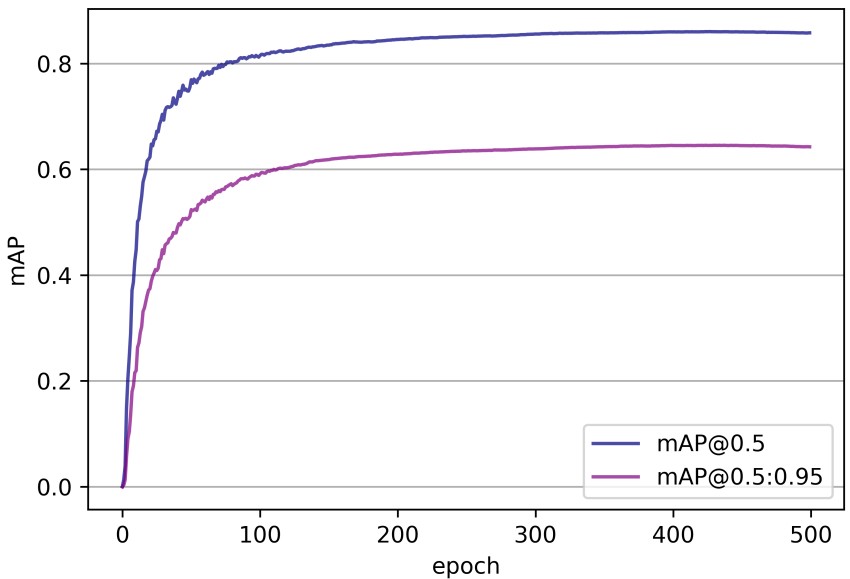

**Figure 13.** Training and validation $mAP$ of the DSDL-Net on the DIOR dataset.

Table 5 presents the time required for training and validation per epoch, measured in minutes, as well as the GPU VRAM consumption during the process. The training set consists of 5862 images with a total of 32,591 instances, while the validation set consists of 5863 images with a total of 35,437 instances.

**Table 5.** Training and validation time and GPU VRAM usage for one epoch.

| Attribute | Value |
| --- | --- |
| Training Time (minute) | 2.87 |
| Validation Time (minute) | 1.1 |
| GPU VRAM Usage (GB) | 8.5 |

## 5. Conclusions

To address the challenges of inaccurate localization and identification in remote sensing object detection, we propose a novel detection network named DSDL. Our network design enables the learning of rich detail information while preserving high-quality semantic features, effectively decoupling the learning of semantics and details within the network. Firstly, we introduce an MRF-DM structure for use within the backbone, capable of retaining and compressing detail features while simultaneously learning high-quality semantic information. Additionally, we propose a CSDF structure for the seamless integration of semantic and detail information at the final stage. It concurrently executes two adaptive learning processes under shared global attention conditions: global and local attention.

Our experimental results demonstrate that the DSDL network exhibits robustness and generalization capabilities when handling remote sensing object detection tasks in complex backgrounds. Its $mAP@0.5$ on the DIOR dataset reached 79.74%, representing a 2.19% improvement over the baseline model, with a parameter reduction of approximately 14%. The network exhibits high precision when detecting categories such as airplanes, basketball

courts, ships, and tennis courts, which commonly appear in complex remote sensing image backgrounds, including urban buildings, bodies of water, and sports venues. However, there remains room for improvement in detecting certain categories, such as bridges, ports, overpasses, train stations, and vehicles. These performance discrepancies may be attributed to the unique characteristics of these categories within remote sensing images.

In summary, our proposed DSDL network effectively enhances localization and identification accuracy in remote sensing object detection within complex background scenarios. This is achieved by introducing MRF-DM and CSDF structures to enable the preservation and fusion of semantic and detail information, allowing for collaborative learning between semantics and details.

**Author Contributions:** Conceptualization, H.R. and W.Q.; methodology, H.R. and Z.Z.; software, H.R. and Y.P.; validation, H.R. and Z.Z.; writing—original draft preparation, H.R.; writing—review and editing, H.R. and W.Q. All authors have read and agreed to the published version of the manuscript.

**Funding:** This work was supported in part by the National Natural Science Foundation of China under Grant 62262028, the National Key Research and Development Program of China under Grant 2022YFD1600202, and the Natural Science Foundation of Jiangxi Province under Grant 20224BAB202020.

**Data Availability Statement:** All datasets used for training and evaluating the performance of our proposed approach are publicly available and can be accessed from [36–38].

**Conflicts of Interest:** The authors declare no conflict of interest.

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
