# Peer review of "A Decoupled Semantic–Detail Learning Network for Remote Sensing Object Detection in Complex Backgrounds"

_electronics, doi:10.3390/electronics12143201_

Round 1

Reviewer 1 Report

The paper proposes a decoupled semantic-detail learning network (DSDL-Net) for detecting multi-scale objects in complex remote sensing backgrounds. It incorporates a multi-receptive field feature fusion and detail mining module to learn higher semantic-level representations and preserve detail texture information. Additionally, an adaptive cross-level semantic-detail fusion network leverages a feature pyramid to fuse detailed and high-level semantic features. Experimental results demonstrate the approach's superiority over 12 benchmark models, achieving improved average precision on three remote sensing datasets.

1.    The authors propose a novel detection network, DSDL, to address the challenges of inaccurate localization and identification in remote sensing object detection. The approach seems promising and tackles an important problem in the field.

2.    The introduction of the MRF-DM structure within the backbone is interesting. It is claimed to retain and compress detail features while learning high-quality semantic information. I would appreciate further clarification on the specific mechanisms employed in this structure and how it achieves the desired objectives.

3.    The CSDF structure for the integration of semantic and detailed information appears to be a key component of the proposed network. The concurrent execution of global and local attention processes under shared global attention conditions is intriguing. Add more details on how these processes are implemented and their impact on the network’s overall performance.

4.    The authors mention certain categories, including bridges, ports, overpasses, train stations, and vehicles, where the performance of the proposed network exhibits room for improvement. It would be valuable to discuss the specific challenges associated with detecting these categories and propose potential avenues for addressing these limitations in future work.

5.    It would also be helpful if the authors could provide insights into the computational requirements and training time of the DSDL network, as these factors can significantly impact the practical feasibility of the proposed approach.

Overall, the paper presents a promising approach for improving localization and identification accuracy in remote sensing object detection. Addressing the above-mentioned points will strengthen the manuscript and contribute to the advancement of the field.

The paper is well-written and organized and the English requires minor editing. 

Reviewer 2 Report

It is very important to have new technology to detect object in complex backgrounds using AI/ML technology. In this paper, combination of various method is key, I believe. In that case, I have several comments and suggestion for your paper to improve.

1)    Figure 1, it is overview structure of DSDL-net but it is very difficult to understand overall outline of your proposed new method with MRFF-DM and CSDF. It seems that you also break down figure 2, 4, 5, 6 and 7. For instance, in figure 1, it looks DM and MRFF are parallel processing but it looks MRFF as first process and then DM as second process. And then, in figure 4, suddenly propose new A-SPPCSP and there is no A-SPPCSP in figure 2. In Figure 5, ACMix (ACmix Attention Module) is also proposed and no explanation in Figure 4.

2)    Ablation experiment metrics, to confirm the accuracy, why you propose to use AP? Usually, to detect objective, there is no average precision (AP) using remote sensing data but it should be truth or false. Of course,  formulation (22)-(24) are important to evaluate the accuracy.

3)    In Figure 8 and 9, since your proposed method is to detect object in  complex backgrounds, do you compare between simple backgrounds and complex backgrounds to convince your new method. In addition, in figure 8, it is difficult to understand legend meaning. Because it is highly depended on accuracy with target object size and density. Can you please explain more detail about the test data and it’s condition in your experiment in Ch 4.4. more clearly in detail?

Result information is too poor. 

Round 2

Reviewer 2 Report

No futher comment

N/A